# Fyn Phosphorylates Transglutaminase 2 (Tgm2) and Modulates Autophagy and p53 Expression in the Development of Diabetic Kidney Disease

**DOI:** 10.3390/cells12081197

**Published:** 2023-04-20

**Authors:** Ryota Uehara, Eijiro Yamada, Shuichi Okada, Claire C. Bastie, Akito Maeshima, Hidekazu Ikeuchi, Kazuhiko Horiguchi, Masanobu Yamada

**Affiliations:** 1Department of Internal Medicine, Division of Endocrinology and Metabolism, Gunma University Graduate School of Medicine, 3-39-15, Showa, Maebashi 371-8511, Japan; 2Division of Biomedical Sciences, Warwick Medical School, Coventry CV4 7AL, UK; 3Department of Nephrology and Hypertension, Saitama Medical Center, Saitama Medical University, 1981 Kamoda, Kawagoe 350-1298, Japan; 4Department of Nephrology and Rheumatology, Gunma University Graduate School of Medicine, Maebashi 371-8511, Japan

**Keywords:** autophagy, diabetic, end-stage renal disease, hyperglycemia, kidney

## Abstract

Autophagy is involved in the development of diabetic kidney disease (DKD), the leading cause of end-stage renal disease. The Fyn tyrosine kinase (Fyn) suppresses autophagy in the muscle. However, its role in kidney autophagic processes is unclear. Here, we examined the role of Fyn kinase in autophagy in proximal renal tubules both in vivo and in vitro. Phospho-proteomic analysis revealed that transglutaminase 2 (Tgm2), a protein involved in the degradation of p53 in the autophagosome, is phosphorylated on tyrosine 369 (Y369) by Fyn. Interestingly, we found that Fyn-dependent phosphorylation of Tgm2 regulates autophagy in proximal renal tubules in vitro, and that p53 expression is decreased upon autophagy in Tgm2-knockdown proximal renal tubule cell models. Using streptozocin (STZ)-induced hyperglycemic mice, we confirmed that Fyn regulated autophagy and mediated p53 expression via Tgm2. Taken together, these data provide a molecular basis for the role of the Fyn–Tgm2–p53 axis in the development of DKD.

## 1. Introduction

One of the major complications of diabetes is diabetic nephropathy (DN), which is a leading cause of end-stage renal disease (ESRD) [1]. Glomerular disorders due to hyperglycemia have been considered to be the main cause of DN. However, in recent years, correlation between renal tubule disorders and DN due to aging or arteriosclerotic lesions has also been suggested [2,3]. Thus, DN is also called diabetic kidney disease (DKD), as the disorder may not be limited to the nephron unit itself. Moreover, autophagy has attracted attention as it may play a role in the development of DKD [2]. While autophagy is a system of bulk degradation, it also maintains intracellular quality in the basal state. Recent studies have demonstrated the role of autophagy in metabolic functions [4,5,6,7,8,9,10], and molecular mechanisms underlining the crosstalk between autophagy and energy metabolism have come under scrutiny. In glomeruli and tubular cells affected by DN, autophagy deteriorates cell quality and causes cell death [2]; however, the mechanism of this is still to be elucidated.

Fyn is a member of the Src family of non-receptor tyrosine kinases and a regulator of T cell signaling in the immune response [11]. Previous studies using Fyn-knockout mice (FynKO) and tissue-specific Fyn-overexpressing mice have demonstrated the important role that Fyn plays in the regulation of insulin-mediated metabolism and in autophagic processes. In particular, Fyn kinase decreases autophagy in the skeletal muscle, a mechanism linked to the development of sarcopenia [12,13].

A phosphotyrosine proteomic screening comparing skeletal muscles from FynKO mice with those from the muscle-specific Fyn-overexpressing mice (HSA-Fyn transgenic mice) identified signal transducer and activator of transcription 3 (STAT3) as a predominant substrate for Fyn kinase [14]. Whilst we demonstrated that Fyn enhances STAT3 phosphorylation and alters the expression of Vps34, as well as the assembly of the Vps34/Vps15/Beclin1/Atg14 complex, the molecular mechanism underlining the regulation of autophagy is still unknown [14]. 

The mechanisms by which Beclin1 complexes regulate autophagy are not well defined. However, studies have suggested that Beclin1 crosslinking by transglutaminase 2 (Tgm2) could be one of the mechanisms of autophagy regulation, resulting in decreased Beclin1 activity and autophagy [15]. More recently, it was reported that Tgm2 binds to the autophagy-specific substrate p62 as well as the tumor suppressor p53 to not only regulate autophagic activity, but also the expression of p53 itself, in tumor cells [16]. Although p53 is an exacerbating factor in the development of DKD, very little is known about the regulation of p53 in the renal context and in autophagic processes [17,18,19]. In this study, we investigated the possible connection between various factors (i.e., Tgm2, p53, and Fyn) otherwise independently linked to autophagy. We found that Fyn kinase plays an important role in regulating autophagy in the kidney, and that the mechanism involves Tgm2 and p53.

## 2. Materials and Methods

### 2.1. Antibodies and Reagents

Rabbit polyclonal antibody against Fyn and mouse monoclonal antibodies against Tgm2, p53, and AQP-1 were purchased from Santa Cruz Biotechnology (Santa Cruz, CA, USA). Rabbit polyclonal antibody against p62 was purchased from Enzo Life Sciences (Farmingdale, NY, USA). Rabbit polyclonal antibody against LC3 was obtained from Cell Signaling Technology (Danvers, MA, USA). Mouse monoclonal antibody against anti-GAPDH antibody was purchased from MBL International (Woburn, MA, USA). The anti-phosphotyrosine clone 4G10 antibody was purchased from Millipore (Billerica, MA, USA). Leupeptin hemisulfate was obtained from ThermoFisher Scientific (Pittsburgh, PA, USA). All other reagents were purchased from Merck (Burlington, MA, USA).

### 2.2. cDNA Constructs

The pcDNA3 and pcDNA3-Fyn-CA-V5 plasmids were constructed as previously published [16]. The IMAGE clone 3256943 (GenBank: BC016492) for mouse Tgm2 was obtained from imaGenes (Berlin, Germany) and PCR was performed with a pair of oligonucleotides (5′-CACCATGGACTACAAGGACGATGACGACAAG-ATGGCAGAGGAGCTG-3′ and 5′-TTAGGCCGGGCCGATGATAA-3′). The PCR product was separated on a 2% agarose gel and the specific single band was extracted using the QIAquick PCR purification kit (Qiagen, Venlo, The Netherlands). The purified PCR product was cloned into pcDNA3.1D/V5-His-TOPO using the pcDNA3.1 Directional TOPO Expression Kit (ThermoFisher). The pcDNA3.1-Flag-Tgm2-Y369 construct was obtained using the QuickChange 2-XL Site-Directed Mutagenesis Kit (Agilent Technologies, Santa Clara, CA, USA) with a pair of oligonucleotides (5′-GAAGAGCGAAGGGACATTCTGTTGTGGCCCA-3′ and 5′-TGGGCCACAACAGAATGTCCCTTCGCTCTTC-3′).

### 2.3. Cell Culture

The human renal proximal tubular epithelial cell line HK-2 was obtained from ATCC (Manassas, VA, USA). The cells were cultured in 100 cm^2^ dishes and grown at 37 °C in a 5% CO_2_ atmosphere in Keratinocyte-SFM medium (Thermo Fisher Scientific, Waltham, MA, USA) supplemented with bovine pituitary extract (0.05 mg/mL) and epithelial growth factor (5 ng/mL; Thermo Fisher Scientific). Cells were sub-cultured at 70–80% confluency. Human renal proximal tubule epithelial cells (PTECs) obtained from ATCC were cultured in 100 cm^2^ dishes and grown at 37 °C in a 5% CO_2_ atmosphere in REBM Basal Medium and REGM SingleQuots supplements as previously described (Lonza Ltd., Muenchensteinerstrasse, Basel, Switzerland).

### 2.4. Transfection

After seeding 2 × 10^5^ cells onto 12-well plates in Dulbecco’s Modified Eagle medium (DMEM) low glucose for 12 h, cells were transfected with either 2 μg of myc-DDK-Tgm2 wild-type (WT) or Tgm2-YF mutants with 2 μL of X-tremeGENE HP DNA transfection reagent (Roche Diagnostics) using OptiMEM (Thermo Fisher Scientific) according to the manufacturer’s protocol. OptiMEM was changed to DMEM low glucose medium after 12 h, and transfected cells were harvested after 48 h. For siRNA-mediated knockdown of Fyn and Tgm2, cells were transfected with siGENOME siRNA SMART pool using DharmafectDuo (Dharmacon, Thermo Scientific, Waltham, MA, USA) according to the manufacturer protocol. Briefly, 16 h after seeding 1.0 × 10^5^ cells onto 12-well plates, cells were transfected with either 0.2 μM of non-target siRNA or Fyn/Tgm2 siRNA with DharmafectDuo followed by harvesting for subsequent experiments after 48 h.

### 2.5. Western Blot Analysis

Proteins were extracted using a lysis buffer containing a protease inhibitor as previously described [16] and quantified using a bicinchoninic acid assay (BCA) protein assay kit (Pierce, Rockford, IL, USA). Equal amounts of protein were separated using 10% sodium dodecyl sulfate-polyacrylamide gel electrophoresis (SDS-PAGE) and transferred onto polyvinylidene fluoride (PVDF) membranes, which were blocked with 5% fatty acid-free powdered milk for 2 h. The membranes were incubated with the following primary antibodies overnight at 4 °C: anti-Fyn (1:1000), anti-p62 (1:1000), anti-LC3 (1:1000), anti-Tgm2 (1:1000), anti-phosphotyrosine (4G10; 1:1000), anti-p53 (1:1000), and anti-GAPDH (1:1000). Thereafter, they were incubated with a horseradish peroxidase-conjugated secondary antibody for 30 min. The membranes were extensively washed in tris-buffered saline with Tween 20, and antigen–antibody complexes were visualized via chemiluminescence using an enhanced luminol-based chemiluminescent (ECL) kit (Pierce).

### 2.6. Immunoprecipitation

Cells were lysed or tissues from mouse models were homogenized using a bead-based homogenizer (bead crusher (μT-12, Taitec, Saitama, Japan) with Zirconia beads (setting: 2500 rpm, 90 s)) in NP-40 lysis buffer. Homogenates were centrifuged at maximum speed (15,000 rpm) for 10 min at 4 °C. Supernatants were collected and protein concentration was determined using the BCA method. Next, 10 mg of the lysate was mixed with 30 μL of anti-Tgm2 antibody while gently rocking for 2 h at 4 °C, and 70 μL of protein A/G PLUS-Agarose was added (Santa Cruz) for 1 h at 4 °C. Samples were washed 3×, boiled, and proteins were separated on a 10% SDS-PAGE and transferred onto PVDF membranes before immunoblotting was performed using the indicated antibodies.

### 2.7. In Vitro Tgm2 Phosphorylation Assay

Human glutathione S-transferase (GST)-Tgm2 fusion protein and Flag-Fyn-CA were purchased from Abnova (Taipei, Taiwan) and BPS Bioscience (San Diego, CA, USA), respectively. GST-Tgm2 (100 ng) was incubated with the active recombinant Fyn kinase (1.8 U) in the presence of ATP and kinase buffer (Cell Signaling) for 1 h at 30 °C. Samples were separated on 8% SDS-polyacrylamide gels and immunoblotting was performed with the 4G10 anti-phosphotyrosine antibody.

### 2.8. Measurement of Autophagy Flux in HK-2/PTECs Cells

Cells were maintained in standard media for 48 h before stimulation with 2 M NH_4_Cl and 10 mM Leupeptin for 2 h. Cells were then harvested and the expression of both LC3 forms (LC3-I and LC3-II) was detected by immunoblotting. 

### 2.9. Animals

The pp59fyn-null C57BL/6 mice (FynKO) were provided by RIKEN BRC with the support of the National BioResource Project of Ministry of Education, Culture, Sports, Science, and Technology, Japan. Heterozygous Fyn mice were bred to produce FynKO and their controls. C57BL/6J mice were purchased from Charles River (Wilmington, MA, USA). All mice were housed in a facility with a 12/12 h light/dark cycle and fed a standard chow diet (Research Diets, New Brunswick, NJ, USA) containing 67% (Kcal) carbohydrates, 19% protein, and 4% fat. All studies were approved by and performed in compliance with the guidelines of the Institutional Animal Care and Use Committee of Gunma University. For the streptozocin (STZ)-induced diabetes studies, animals (8–12-week-old males) were injected with STZ (50 mg/kg) dissolved in citrate buffer (pH 4.5), or the same amount of citrate buffer for controls, intraperitoneally. The injection was administered twice every 5 days consecutively for two weeks. Animals (maintained on a standard chow diet) were observed for another two months for the establishment of continuous hyperglycemia. Mice with fasting glycemia above 300 mg/dL were considered diabetic and used for experiments. The mice were then fasted for 16 h, sacrificed via cervical dislocation, and kidneys were collected, according to the protocol adopted by the Animal Models of Diabetic Complications Consortium [20].

### 2.10. Immunofluorescence

Kidneys were dissected and frozen tissue slides were blocked in 1% bovine serum albumin (BSA) at room temperature for 30 min followed by incubation with the diluted primary antibody (1:50 for p62 and AQP-1) in 1% BSA at room temperature for 60 min. The secondary antibodies used were the Alexa Fluor 488 anti-goat and Alexa Fluor 546 anti-rabbit antibodies (Thermo Fisher Scientific). Sections were washed thrice with phosphate-buffered saline (PBS), mounted with Prolong Gold antifade reagent with 4′,6-diamidino-2-phenylindole (DAPI; Thermo Fisher Scientific), and imaged using a confocal fluorescence microscope, BZ-X710 (KEYENCE, Osaka, Japan). Settings (Iris (pinhole), laser intensity, gain, and offset) were fixed for all samples. The p62 signal was quantified using Image J software (National Institutes of Health, New York, NY, USA). Images of 15 representative cells were processed and data represent three independent experiments.

### 2.11. Statistics

Results are expressed as the mean ± standard error of the mean (s.e.m.). Differences between animals and/or treatments were evaluated for statistical significance using the Student’s unpaired *t*-test.

## 3. Results

### 3.1. Fyn Kinase Inhibits Autophagy in HK-2 Cells

Fyn has recently been described as a potential target in acute kidney injury [21]. This strongly suggests that Fyn may be involved in the pathophysiology of kidney disease, in which autophagy is thought to be a possible mechanism. Notably, Fyn plays an active role in the development of sarcopenia by decreasing the Vps34/p150/Beclin1/Atg14 complex and consequently inhibiting macroautophagy in the skeletal muscle [14]. Therefore, we speculated that Fyn expression might also impact autophagy in the kidney. 

To examine this possibility, Fyn expression was significantly reduced in human kidney HK2 cells using siRNA technology, and autophagy flux was assessed. Autophagy is a highly dynamic process, and the sole determinant of steady-state levels of markers such as LC3 (a component of the autophagosome membrane) gives little information about the amplitude of autophagy processes. LC3 is initially synthesized in an unprocessed form, proLC3, that is converted into a proteolytically processed form, LC3-I, which is finally modified into the PE-conjugated form, LC3-II. LC3-II is the only protein marker that is reliably associated with completed autophagosomes, but it is also localized to phagophores. However, the total levels of LC3 do not necessarily change in a predictable manner, as there may be increases in the conversion of LC3-I to LC3-II, or a decrease in LC3-II relative to LC3-I if degradation of LC3-II via lysosomal turnover is particularly rapid [22]. Moreover, the pattern of LC3-I to LC3-II conversion seems not only to be cell-specific, but also related to the kind of stress to which cells are subjected [22]. Therefore, to accurately assess the autophagy flux, ammonium chloride and Leupeptin (NH_4_Cl/Leupeptin) are commonly used to block the turn-over of these markers [22]. NH_4_Cl/Leupeptin (N/L) exposure (Figure 1A) resulted in a significant increase in LC3-II protein levels in control cells, demonstrating that autophagy was efficiently blocked. However, the autophagic flux was greatly enhanced in Fyn-knockdown HK2 cells in response to N/L treatment, as demonstrated by a nearly five-fold increase in LC3-II protein levels (Figure 1A,B). Similarly, there was a 50% reduction in the levels of the well-characterized autophagy substrate p62 in Fyn-knockdown cells (Figure 1C,D), which suggested that a reduction in Fyn expression in HK2 cells resulted in increased autophagy.

### 3.2. Transglutaminase 2 (Tgm2) Regulates Autophagy Processes in Kidney Cells

Transglutaminase 2 (Tgm2) plays an important role in the maturation of autophagosomes [15], and ablation of Tgm2 leads to the accumulation of LC3-II proteins in the heart of Tgm2-knockout mice [23]. Notably, reduction of Tgm2 using siRNA technology in HK2 cells increased LC3-II levels after N/L treatment (Figure 2A,B) and decreased p62 protein expression levels (Figure 2C,D), suggesting that a lack of Tgm2 also induced an increase in autophagy processes in kidney cells. Consistent with this, N/L treatment in the overexpression of Flag-tagged Tgm2 (Flag-Tgm2) cells did not induce LC3-II accumulation, while control cells expressing the empty vector had a persistent autophagy flux (Figure 2E,F).

### 3.3. Fyn-Specific Tgm2 Phosphorylation on Tyrosine 369 (Y369) Impacts Autophagy Processes

Our previously published phosphotyrosine proteomics using Fyn-knockout mice and muscle-specific Fyn-overexpressing transgenic mice (HSA-Fyn) [14] revealed high levels of Tgm2 phosphorylation on two tyrosine residues (Appendix A) in HSA-Fyn mice. Thus, we speculated that Fyn might phosphorylate Tgm2. To investigate this further, HEK293 cells were co-transfected with a V5 epitope-tagged constitutionally active form of Fyn (V5-Fyn-CA) and Flag-Tgm2. Exogenous Flag-Tgm2 was immunoprecipitated using a Flag antibody and total phosphorylation of Flag-Tgm2 was examined using the 4G10 phosphotyrosine antibody. As apparent in Figure 3A (right panel), total Tgm2 phosphorylation levels increased in presence of Fyn-CA. In addition, in vitro kinase assays demonstrated that purified GST-Tgm2 fusion protein was phosphorylated by purified recombinant constitutionally active Fyn kinase (Figure 3B), strongly supporting our hypothesis that Tgm2 is a substrate for the Fyn kinase. 

We identified two putative sites for Fyn phosphorylation, Y369 and Y617 (Appendix A), and generated double-point mutants where Y369 and Y617 were substituted with phenylalanine residues. Co-expression of V5-Fyn-CA with native (WT) or double mutant (D-YF) Tgm2 demonstrated equal expression levels for all forms of Tgm2 (Appendix A), demonstrating that the mutations had no effect on protein expression. However, total phosphorylation Tgm2 levels were dramatically reduced by the double mutation, strongly suggesting that Y369 and/or Y617 may be specific for Fyn phosphorylation (Appendix A, right panel). 

Single mutations were generated and phosphorylation levels of each single mutant were assessed in presence of V5-Fyn-CA. Figure 3C (right panel) shows that the Y369F mutation in Tgm2 significantly reduced phosphorylation levels of Tgm2 in the presence of the Fyn kinase. Interestingly, we observed that the single Y369F mutation reduced Tgm2 phosphorylation to similar levels as those observed in the double D-YF mutant, suggesting that the Y617F mutation had minor effects on Fyn-induced Tgm2 phosphorylation. This also agreed with our phosphotyrosine proteomic screen showing that phosphorylation on Y369 was ~50 times more abundant than that of Y617 (Appendix A), and with the phosphorylation-predicting Netphos3.1 software giving a higher score for Y369 than Y617 in mice (Appendix A). Importantly, Y617 was not predicted to be a potential site in humans (Appendix A).

As such, we focused on the Y369F Tgm2 mutant and examined its effects on autophagy. HK2 cells were transfected with either Flag-tagged Tgm2 native form (WT) or Y369F Tgm2 mutant before being treated with (N/L) or without (C) NH_4_Cl/Leupeptin for 2 h. LC3-II accumulation after NH_4_Cl/Leupeptin treatment was significantly increased (two-fold) when cells were transfected with the Y369F Tgm2 mutant (Figure 3D,E), suggesting that the mutation enhanced autophagy. Consistent with Figure 2E, we did not observe increased autophagy flux in cells overexpressing Tgm2 (Tgm2-WT), confirming that Tgm2 acts as a dominant-negative regulator of autophagy that was de-repressed when it was mutated on Y369.

### 3.4. Tgm2 Knockdown Regulates p53 through Autophagy in Kidney Cells

Both Fyn and Tgm2 have been demonstrated to interact with Beclin1, an important factor in the regulation of autophagy [14,15]. In addition, Tgm2 crosslinking with Beclin1 results in decreased Beclin1 activity and reduced autophagy [15]. Therefore, we first examined whether the crosslinking between Tgm2 and Beclin1 might drive the autophagy processes in kidney cells. We found no evidence of Beclin1 crosslinking with Tgm2 in HK-2 cells transfected with either the empty vector or the V5-Fyn-CA construct (Appendix A). We also did not find any crosslinking in cells overexpressing either the Tgm2 native form (WT) or the Y369F Tgm2 mutant (Appendix A), suggesting that another mechanism might be taking place.

High glucose levels increase the expression of the tumor suppressor p53, and several signaling pathways involving p53 have been associated with the establishment of DKD [24,25,26,27] More recently, it was reported that Tgm2 binds to both p62 and p53, regulating p53 through autophagy in tumor cells [16].

Since HK-2 cells were immortalized cells and p53, a tumor suppressor gene, might not function physiologically [28], we took advantage of primary renal tubule epithelial cells (PTECs) to examine the possible interaction between Tgm2 and p53. We found that Tgm2 was largely expressed in PTECs, and expression was greatly reduced in cells transfected with Tgm2 siRNA (Figure 4A). Consistent with what we observed in siTmg2-HK2 cells (Figure 2), p62 expression was also decreased in siTgm2-PTECs (Figure 4B, left panel). In addition, p53 protein levels were approximately decreased by 50% (Figure 4A,B, right panel). Importantly, we observed a two-fold increase in p53 accumulation after N/L treatment in non-target cells, showing that p53 is an autophagy substrate in PTECs. Consistent with the increased autophagy observed in Tgm2-knockdown HK2 cells, the fold change in p53 protein levels after N/L treatment was much higher in siTgm2 PTECs, demonstrating Tgm2-regulated autophagy-mediated p53 expression.

### 3.5. FynKO Rescues STZ-Induced DKD by Inducing Autophagy-Mediated p53 Degradation

As we demonstrated that mutation of Tgm2 on Fyn-specific Y369 impacted autophagy flux in vitro, we examined Tgm2 phosphorylation levels and their impact on autophagy processes in vivo by taking advantage of the Fyn-knockout (FynKO) mouse model.

It has been widely demonstrated that DKD is correlated with obesity-induced hyperglycemia [29,30]. However, whilst FynKO mice become obese after 8–10 weeks on a 60% Kcal high-fat diet (HFD), their glycemia remains similar to that of control standard chow diet-fed mice [30]. Therefore, to efficiently induce hyperglycemia in both control and FynKO mice, animals were treated with STZ, a glucosamine–nitrosourea compound that damages pancreatic beta cells and induces diabetes [31]. Similar hyperglycemia levels were observed in both control (WT) and FynKO mice at 2 weeks of STZ treatment (Figure 5A), and mice were maintained on a standard chow diet for an additional 8 weeks, a timeframe long enough to trigger DKD [32]. Kidneys were collected and slides were prepared to examine signs of autophagy via visualization of p62 puncta using immunohistofluorescence in both WT and FynKO mice treated or not with STZ. As apparent in Figure 5B,C and Appendix A, p62 signal intensity increased significantly in the kidney renal tubular cells of STZ-induced diabetic WT mice, indicating a lack of autophagy in renal tubular cells. 

However, no change was observed in the renal tubular cells of STZ-induced diabetic FynKO mice, suggesting that the lack of Fyn protected against diabetes-driven inhibition of autophagy (Figure 5B,C). Consistent with our in vitro data, p53 expression increased in the kidneys of STZ-induced diabetic WT mice but not in the kidneys of diabetic FynKO mice, suggesting that Fyn regulated autophagy and p53 expression in vivo (Figure 5D,E).

## 4. Discussion

The data presented in this study provide a novel Beclin1- and hyperglycemia-independent autophagy mechanism in DKD. We identified Fyn as a novel autophagy regulator in proximal tubular cells.

DKD is a chronic kidney disease caused by hyperglycemia, induced by many factors such as accumulation of advanced glycation products (AGEs), free radicals, or activation of protein kinase C [33]. More recently, a variety of studies established the novel concept that autophagy also has important roles in the pathogenesis of chronic kidney disease [2,27]. However, the role of autophagy in DKD remained elusive [34].

Proximal tubular cells are responsible for the reabsorption of nutrients and consume a large amount of energy [35]. They are susceptible to metabolic changes and have higher autophagic activity than other kidney cells. Their dysfunction may be closely related to the progression of DKD [35]. Hyperglycemia reportedly inhibits the expression of Beclin1, resulting in the inhibition of autophagosome membrane formation [36], but the underlying mechanisms of how Beclin1 modulates autophagy in proximal tubular cells under diabetic conditions are still elusive [35].

Tgm2 is a member of a family of Ca^2+^-dependent transamidases, catalyzing the isopeptide bond between glutamate and lysine residues, resulting in a covalent crosslink [37]. To date, discrete roles of Tgm2 have been identified [38,39,40,41]. A recent study showed that the role of extracellular Tgm2 is induced in glucose-stimulated cell lines, and that Y369 is a potential phosphorylation site [24]. However, the role of Tgm2 in renal tubular cells under diabetic conditions was unclear and the significance of tyrosine phosphorylation or the tyrosine kinase responsible for the Tgm2 phosphorylation was still unknown [24]. Here, we identified Tgm2 as a novel autophagy regulator in proximal tubular cells. We identified Fyn as the kinase that phosphorylates Tgm2 at Y369 and showed that this phosphorylation determined the autophagic activity in proximal tubular cells.

It was reported that Beclin1 is crosslinked with Tgm2 [15]. Therefore, we first examined if Beclin1 crosslinking with Tgm2 also occurred in HK-2 cells; however, this was not the case. A more recent study showed that Tgm2 forms a complex with p53 and p62, known autophagy regulators/substrates, to degrade p53 in autophagosomes in cancer cells [16]. p53, a well-known tumor suppressor, is a key component of the cellular response to stress [25]. Indeed, p53 knockdown in renal proximal tubules protects against ischemic–reperfusion injury. Importantly, p53 knockdown in other renal tubular segments is ineffective [26]. In the present study, we found that autophagy regulated Tgm2-mediated p53 expression, implicating the role of Tgm2-regulated autophagy. Thus far, it was widely accepted that p53 is degraded via ubiquitination [30]. Therefore, this is the first evidence that p53 is degraded via autophagy in renal proximal tubular cells. Moreover, given that p53 activation in tubular cells plays a critical role in acute kidney injury (AKI) pathogenesis and maladaptive kidney repair after AKI [42], p53 regulation by Tgm2 could be a novel potential target for the treatment of this disorder.

Lastly, we utilized STZ-induced hyperglycemic mice to confirm the underlying mechanisms of Tgm2 phosphorylation by Fyn in regulating autophagy and p53 expression in DKD. We demonstrated that upon STZ-induced DKD, p53 was only decreased in FynKO mice, along with Tgm2 protein expression, therefore providing the first evidence of a molecular basis for the role of the Fyn–Tgm2–p53 axis in the development of DKD.

Interestingly, inhibition of autophagy in kidneys of obese patients has been described [43,44]. In line with this, we found a two-fold increase in p62 puncta numbers in the kidneys of WT mice fed a HFD (60% Kcal) compared with mice fed a standard diet, strongly supporting the inhibition of autophagic processes (Figure 5A,B). Both p53 and Tgm2 protein levels were increased in HFD-fed mice (Figure 5C–F). Interestingly, Tgm2 tyrosine phosphorylation levels were increased two-fold in the kidneys of HFD-fed mice (Figure 5G), which may be correlated with an increase in Fyn kinase enzymatic activity previously reported in this condition [13]. Whilst these data are supportive of a mechanism by which the Fyn kinase regulates autophagy in the obese kidney, tissue-specific or inducible knockout mice for time-dependent gene control are necessary to confirm these mechanisms. Notably, however, this regulatory mechanism does not seem to be mediated by blood glucose levels, and this may lead to novel therapeutic targets for DKD regardless of the control of diabetes.

## Figures and Tables

**Figure 1 cells-12-01197-f001:**
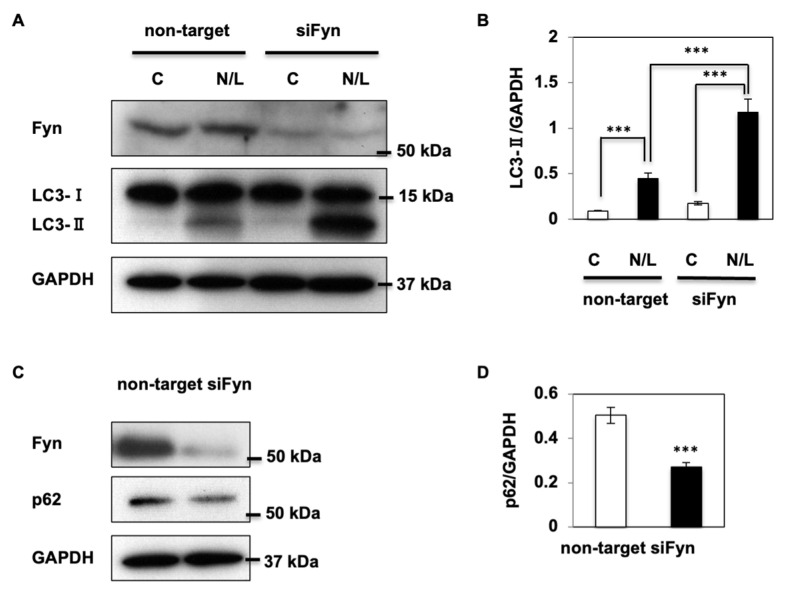
Fyn kinase inhibits autophagy in HK-2 cells. (**A**) HK-2 cells were transfected with non-target or Fyn siRNA to generate Fyn-knockdown (siFyn) or control cells. After 48 h, cells were treated with NH_4_Cl and Leupeptin (N/L) for 2 h, then harvested, and levels of autophagy markers were determined by immunoblotting. Blots are representative images of three independent experiments. (**B**) LC3-II expression was normalized to glyceraldehyde 3-phosphate dehydrogenase (GAPDH) in non-target and siFyn cells in the presence (N/L) or absence (C) of NH_4_Cl and Leupeptin, and signals were quantified. (**C**) p62 protein levels were determined in control cells and siFyn cells using Western blotting. (**D**) p62 protein expression was normalized to GAPDH and signals were quantified. Data are shown as the mean ± standard error of the mean (s.e.m.), *** *p* < 0.001.

**Figure 2 cells-12-01197-f002:**
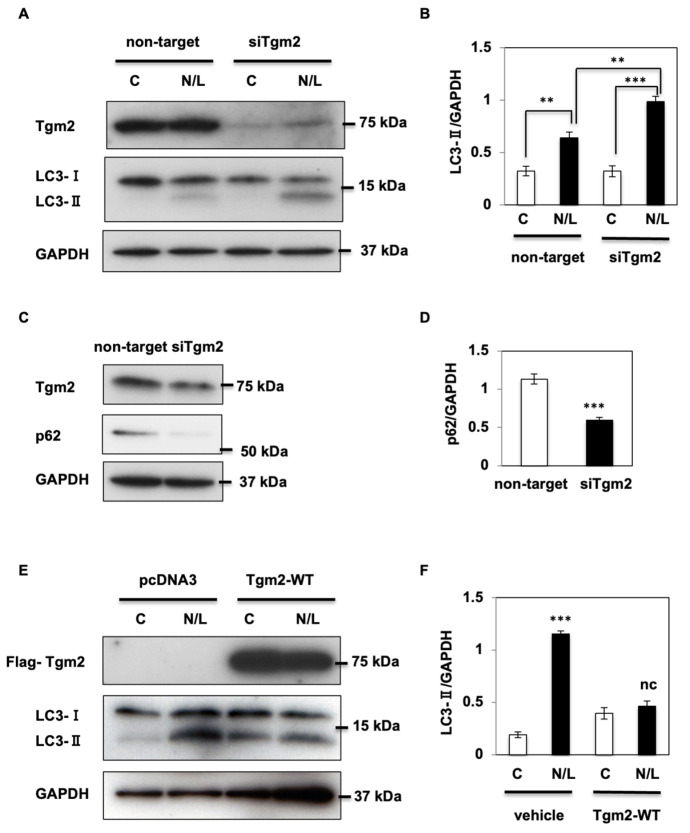
Transglutaminase 2 (Tgm2) regulates autophagy processes in kidney cells. (**A**) Transient knockdown of Tgm2 in HK2 cells was achieved using siRNA technology (siTgm2) and control cells were transfected with non-target RNA. Cells were treated with (N/L) or without (C) NH_4_Cl and Leupeptin for 2 h. Cell lysates were prepared, and Tgm2, LC3-I, and LC3-II protein expression levels were examined using immunoblotting. (**B**) LC3-II signal was corrected using GAPDH levels. (**C**) p62 protein levels were determined using immunoblotting in HK2 cells transfected with siRNA Tgm2 or non-target RNA. (**D**) p62 expression signal was quantified. (**E**) HK2 cells were transfected with a Flag-tagged empty vector (pcDNA) or with a vector expressing Flag-Tgm2 (Tgm2-WT). Tgm2 protein levels were measured using a Flag antibody. LC3-I and LC3-II proteins levels were assessed after a 2 h treatment with NH_4_Cl and Leupeptin. (**F**) LC3-II signal was corrected using GAPDH levels. Data are shown as the mean ± s.e.m., ** *p* < 0.01, *** *p* < 0.001.

**Figure 3 cells-12-01197-f003:**
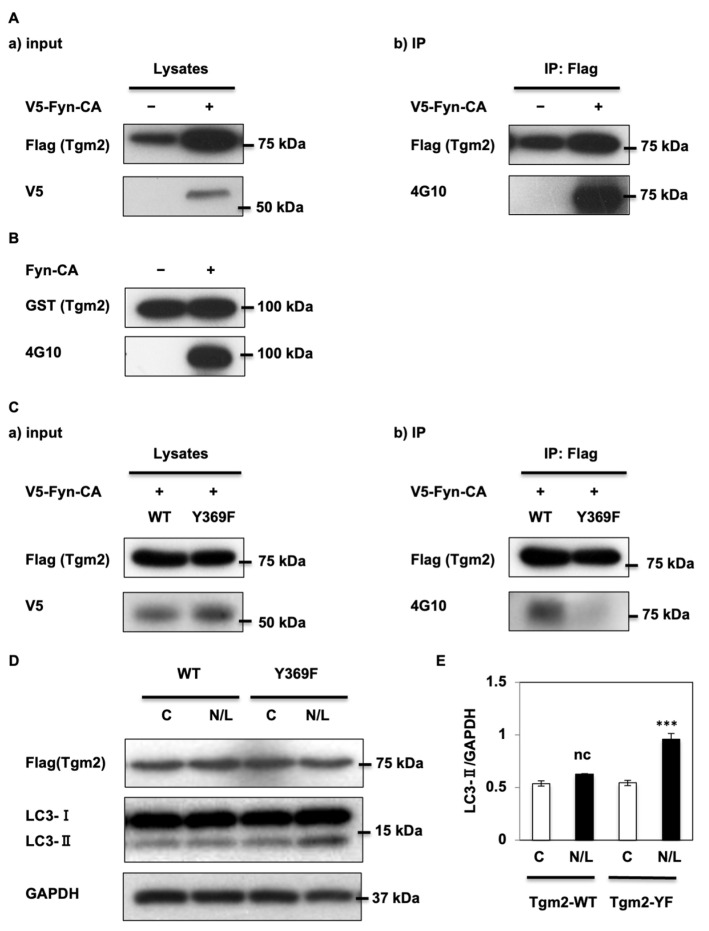
Fyn kinase phosphorylates Tgm2 on the tyrosine 369 (Y369) residue. (**A**) 48 h after HEK-293T cells were co-transfected with Flag-Tgm2 and V5-Fyn-CA, Tgm2 immunoprecipitation was performed using a Flag antibody followed by Western blotting. Blots are representative images of three independent experiments. (**B**) Purified human glutathione S-transferase (GST)-Tgm2 and either mock or Flag-Fyn-CA were incubated at 30 °C with Src kinase buffer supplemented with ATP for 30 min. Western blotting was performed with the indicated antibodies. (**C**) 48 h after HEK-293T cells were co-transfected with V5-Fyn-CA and Flag-Tgm2 or Flag-Tgm2-Y369F, immunoprecipitation was performed using a Flag antibody followed by Western blotting with the indicated antibodies. Blots are representative images of three independent experiments. (**D**) Cells were transfected with the native form (wild-type WT) or the Y369F-Tgm2 mutant (Y369F) and treated with (N/L) or without (C) NH_4_Cl and Leupeptin for 2 h. Tgm2 and LC3 expression levels were determined using immunoblotting and (**E**) the signals were quantified. Data are shown as the mean ± s.e.m., *** *p* < 0.001.

**Figure 4 cells-12-01197-f004:**
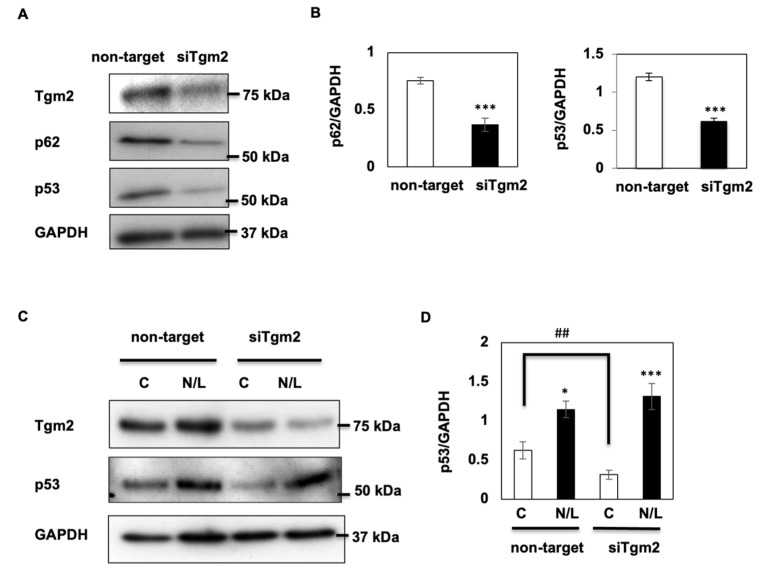
Tgm2 regulates p53 protein expression via autophagy. (**A**) PTECs were transfected with either non-target or Tgm2 siRNA for 48 h. Cells were harvested and expression levels of Tgm2, p62, and p53 proteins were determined using specific antibodies. (**B**) Quantification of p62 and p53 expression levels normalized to GAPDH. (**C**) PTECs transfected with either non-target or Tgm2 siRNA for 48 h and incubated (or not) with NH_4_Cl and Leupeptin for 2 h. Tgm2 and p53 expression levels were determined by Western blotting. Representative images of four independent experiments are shown. (**D**) p53 expression levels normalized to GAPDH. Data are shown as the mean ± s.e.m., * *p* < 0.05, ^##^
*p* < 0.01, *** *p* < 0.001.

**Figure 5 cells-12-01197-f005:**
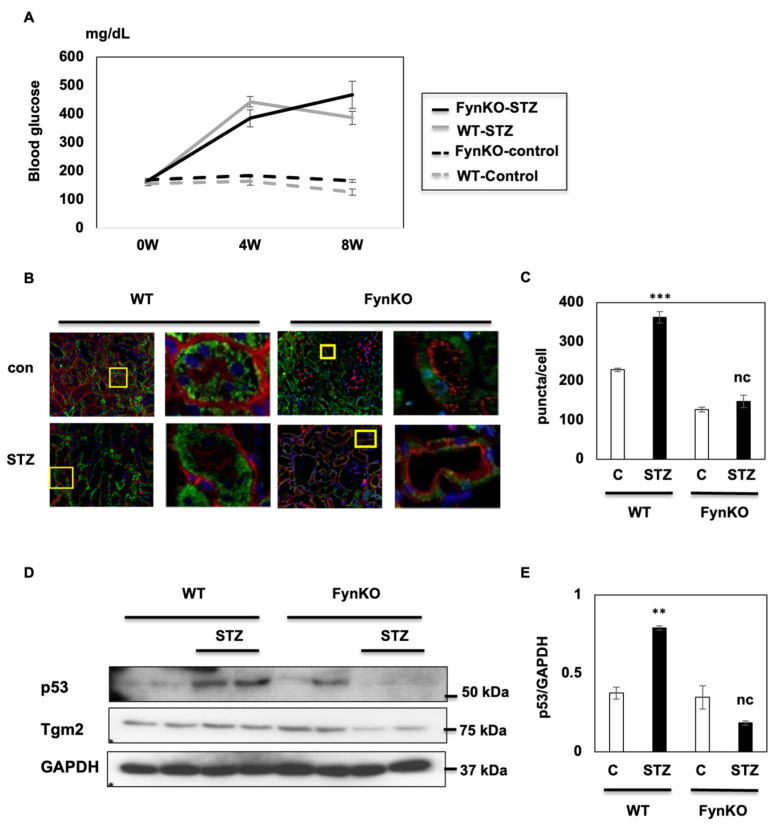
FynKO rescues STZ-induced DKD by inducing autophagy-mediated p53 degradation. (**A**) Fasting blood glucose levels in WT and FynKO mice before and after STZ treatment (**B**) Kidneys were harvested from STZ-treated (and non-treated) FynKO and control mice (WT). Slices were prepared and stained for p62 (green), AQP1 (red), and DAPI (blue). (**C**) Quantification of p62 puncta in each cell described in (**B**). (**D**) p53 protein levels in kidneys of STZ-treated (or not) FynKO and control (WT) mice. These are representative images of three independent experiments. (**E**) p53 expression was normalized to GAPDH. Data are shown as the mean ± s.e.m., ** *p* < 0.01, *** *p* < 0.001.

## Data Availability

Not applicable.

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
