# Peer review of "Fyn Phosphorylates Transglutaminase 2 (Tgm2) and Modulates Autophagy and p53 Expression in the Development of Diabetic Kidney Disease"

_cells, 2023, doi:10.3390/cells12081197_

Round 1

Reviewer 1 Report

The manuscript entitled "Fyn phosphorylates transglutaminase 2 (Tgm2) and modulates autophagy and p53 expression in the development of diabetic kidney disease” is an interesting research work. Design of experimental work is good. The results and discussion both scientifically and technically well written. There are few spelling and grammar mistakes that could be fixed during revision process.

I suggest minor revisions:

Results: Authors should justify the choice of performing the quantitative analysis of LC3 as LC3II/GAPDH and not LC3II/LC3I.

The text also needs to be revised to eliminate some repetitions and “residual parts” of the text. For example delete in Author Contributions: “For research articles with several authors, a short paragraph specifying their individual contributions must be provided. The following statements should be used…”

Author Response

April 8, 2023

Prof. Dr. Cord Brakebusch, Prof. Dr. Alexander E. Kalyuzhny 

Editors-in-Chief

Cells

RE: Fyn phosphorylates transglutaminase 2 (Tgm2) and modulates autophagy and p53 expression in the development of diabetic kidney disease

Dear Dr. Paloma Alonso-Magdalena,

We wish to re-submit the attached manuscript as an Original Article. The manuscript ID is cells-2239541.

The manuscript has been revised and appropriate changes have been made in accordance with the reviewers’ suggestions. The responses to their comments are given below.

We thank the reviewers for the thoughtful suggestions and insights, which have enriched the manuscript and produced a better and more balanced account of the research. We hope that the revised manuscript is now suitable for publication.

Reviewer 1

The manuscript entitled "Fyn phosphorylates transglutaminase 2 (Tgm2) and modulates autophagy and p53 expression in the development of diabetic kidney disease” is an interesting research work. Design of experimental work is good. The results and discussion both scientifically and technically well written. There are few spelling and grammar mistakes that could be fixed during revision process.

I suggest minor revisions:

Results: Authors should justify the choice of performing the quantitative analysis of LC3 as LC3II/GAPDH and not LC3II/LC3I.

- We appreciate the reviewer’s comment regarding very important point. As mentioned in Ref.25, assays for autophagy are often technically challenging to perform. In the revised manuscript, we added cautionary notes in the results section, and we hope this will be helpful for both reviewers and readers: LC3 is initially synthesized in an unprocessed form, proLC3, which is converted into a proteolytically processed form, LC3-I, and is finally modified into a PE-conjugated form, LC3-II. LC3-II is the only protein marker that is reliably associated with complete autophagosomes, but it can also be localized to phagophores. However, the total levels of LC3 do not necessarily change in a predictable manner, as there may be increases in the conversion of LC3-I to LC3-II, or a decrease in LC3-II relative to LC3-I if degradation of LC3-II via lysosomal turnover is particularly rapid [25]. Moreover, the pattern of LC3-I to LC3-II conversion seems not only to be cell specific, but also related to the kind of stress to which cells are subjected [25].

The text also needs to be revised to eliminate some repetitions and “residual parts” of the text. For example delete in Author Contributions: “For research articles with several authors, a short paragraph specifying their individual contributions must be provided. The following statements should be used…”

- Thank you your understanding that English is not the native language of the authors. The manuscript has now been thoroughly revised and amended to correct language, structure and grammatical errors. We hope that it is now adequate for publication.

Reviewer 2 Report

In the paper named ”Fyn phosphorylatestransglutaminase 2 (Tgm2) and modulates autophagy and p53 expression in the  development of diabetic kidney disease” author aim to known the Fyn role in the autophagic process and therefore provide a molecular basis for the role of the Fyn-Tgm2-p53 axis in the development of DKD. In this sense author found that Fyn-dependent phosphorylation of Tgm2 regulates autophagy in proximal renal tubules in vitro. Moreover in these conditions p53 expression is decreased upon autophagy in Tgm2- knockdowned proximal renal tubule cell models.  Therefore we can say that author confirmed that Fyn regulated autophagy and mediated p53 expression via Tgm2 using a streptozocin (STZ)-induced hyperglycemia mice model.

Only minor questions are required

1)     pcDNA plasmid and pcDNA3-Fyn-CA-V5 have both a flag epitope? In the western shown in figure 2E only signal for flag is seen in Tm3WT however this flag must be included in the plasmid

2)     In the Inmunoprecipitation method author use incubations with the antibody only 2h at 4ºC. Habitually this incubation is performed overnight, have author signal only using 2 hours??

3)     Pull down assays is missing in material and methods.

4)     In line 184 author say that they observed the animals (on CT)for two months, please can clarify what means CT? They mean PET/CT?

5)     In figure 2E in the western it is very curiously that the pcDNA3 alone has signal for LC3-I and LC3-II

6)     In figure 3 authors make a Kinase assay using Src Kinase but in material and methods this method is missing

7)     How author identified the Fyn Phosphorilation Y369 and Y617 in the previous Phosphoproteome analysis? In the supplementary figure only the peptide sequence is include. Please in case of the previous work was used to this identification this issue must be clarifying in the text.

8)     In paragraph between lines 320 and 328 authors say that the Y369f mutation is the involved in the phosphorilation reduction but results about the Y617F and Y369F were not shown?

9)     Authors suggest that p53 expression is regulated by Tgm2 via autophagy. Have author make some experiments using Si RNAp53 to reduce P53 and confirm this results?

Author Response

April 15, 2023

Prof. Dr. Cord Brakebusch, Prof. Dr. Alexander E. Kalyuzhny 

Editors-in-Chief

Cells

RE: Fyn phosphorylates transglutaminase 2 (Tgm2) and modulates autophagy and p53 expression in the development of diabetic kidney disease

Dear Dr. Paloma Alonso-Magdalena,

We wish to re-submit the attached manuscript as an Original Article. The manuscript ID is cells-2239541.

The manuscript has been revised and appropriate changes have been made in accordance with the reviewers’ suggestions. The responses to their comments are given below.

We thank the reviewers for the thoughtful suggestions and insights, which have enriched the manuscript and produced a better and more balanced account of the research. We hope that the revised manuscript is now suitable for publication.

Reviewer 2

In the paper named ”Fyn phosphorylatestransglutaminase 2 (Tgm2) and modulates autophagy and p53 expression in the  development of diabetic kidney disease” author aim to known the Fyn role in the autophagic process and therefore provide a molecular basis for the role of the Fyn-Tgm2-p53 axis in the development of DKD. In this sense author found that Fyn-dependent phosphorylation of Tgm2 regulates autophagy in proximal renal tubules in vitro. Moreover in these conditions p53 expression is decreased upon autophagy in Tgm2- knockdowned proximal renal tubule cell models.  Therefore we can say that author confirmed that Fyn regulated autophagy and mediated p53 expression via Tgm2 using a streptozocin (STZ)-induced hyperglycemia mice model.

Only minor questions are required

  • pcDNA plasmid and pcDNA3-Fyn-CA-V5 have both a flag epitope? In the western shown in figure 2E only signal for flag is seen in Tm3WT however this flag must be included in the plasmid

- We thank the reviewer for their careful reading and for pinpointing details that needed to be clarified. Indeed, pcDNA3-Fyn-CA-V5 does not have a flag. In Figure 2E, the pcDNA3 plasmid has a flag but only expressed flag could not detect standard SDS-PAGE. Now amended at legend in 2(E) “HK2 cells were transfected with an empty vector (pcDNA) expressed Flag epitope tag or with a vector expressing Flag-Tgm2 (Tgm2-WT). “

  • In the Inmunoprecipitation method author use incubations with the antibody only 2h at 4ºC. Habitually this incubation is performed overnight, have author signal only using 2 hours??

- The reviewer is correct and most, but not all immunoprecipitations are done O/N. However, we have significant experience using this Flag antibody (cite publications where you also use 2hours). In addition, our preliminary experiments using timings between 2 h and o/N incubation time showed that 2 h was the optimum time as it was significantly reducing the background noise in our setting.

  • Pull down assays is missing in material and methods.

- We have not performed pull down assays for this paper but instead performed immunoprecipitations, which is described in material and methods.   

  • In line 184 author say that they observed the animals (on CT)for two months, please can clarify what means CT? They mean PET/CT?

- We apologize our abbreviation was not mentioned before. “on CD” means “on a standard Chow Diet”. We have now amended this and included the following sentence in our manuscript: Animals were observed for another two months (on a standard Chow Diet) for the establishment of continuous hyperglycemia.

  • In figure 2E in the western it is very curiously that the pcDNA3 alone has signal for LC3-I and LC3-II

- It was indicated that only the empty vector expression did not affect autophagy. We are now mentioned this in the result section: Consistent with this, N/L treatment in the overexpression of Flag-tagged Tgm2 (Flag-Tgm2) cells did not induce LC3-II accumulation while control cells expressing empty vector had a persistent autophagy flux.

  • In figure 3 authors make a Kinase assay using Src Kinase but in material and methods this method is missing

- We respectfully disagree with the reviewer as the method is currently described in 2.7 in the material and method section.

  • How author identified the Fyn Phosphorilation Y369 and Y617 in the previous Phosphoproteome analysis? In the supplementary figure only the peptide sequence is include. Please in case of the previous work was used to this identification this issue must be clarifying in the text.

- The reviewer can easily find this information in 2.7 in Supplementary Material, Figure legend, Figure S1.

  • In paragraph between lines 320 and 328 authors say that the Y369f mutation is the involved in the phosphorilation reduction but results about the Y617F and Y369F were not shown?

- These results are currently displayed in Supplementary Figure 1C.

9)     Authors suggest that p53 expression is regulated by Tgm2 via autophagy. Have author make some experiments using Si RNAp53 to reduce P53 and confirm this results?

- The purpose of our paper was the show that p53 expression is regulated by the axis Fyn-Tgm2, and this implicates autophagy in renal proximal tubular cells. We are not sure how knocking down p53 via siRNA technology would particularly show in this context as if p53 is not expressed, there would obviously not be any regulation via Fyn-Tgm2.